**∂ | Open Peer Review** | Microbial Ecology | Research Article

# The impact of DNA extraction on the quantification of *Legionella*, with implications for ecological studies

Alessio Cavallaro,[1,2] Marco Gabrielli,[1] Frederik Hammes,[1] William J. Rhoads[1]

**ABSTRACT** Monitoring the levels of opportunistic pathogens in drinking water is important to plan interventions and understand the ecological niches that allow them to proliferate. Quantitative PCR is an established alternative to culture methods that can provide a faster, higher-throughput, and more precise enumeration of the bacteria in water samples. However, PCR-based methods are still not routinely applied for *Legionella* monitoring, and techniques, such as DNA extraction, differ notably between laboratories. Here, we quantify the impact that DNA extraction methods had on downstream PCR quantification and community sequencing. Through a community science campaign, we collected 50 water samples and corresponding shower hoses, and compared two commonly used DNA extraction methodologies to the same biofilm and water phase samples. The two methods showed clearly different extraction efficacies, which were reflected in both the quantity of DNA extracted and the concentrations of *Legionella* enumerated in both the matrices. Notably, one method resulted in higher enumeration in nearly all samples by about one order of magnitude and detected *Legionella* in 21 samples that remained undetected by the other method. 16S rRNA amplicon sequencing revealed that the relative abundance of individual taxa, including sequence variants of *Legionella*, significantly varied depending on the extraction method employed. Given the implications of these findings, we advocate for improvement in documentation of the performance of DNA extraction methods used in drinking water to detect and quantify *Legionella*, and characterize the associated microbial community.

**IMPORTANCE** Monitoring for the presence of the waterborne opportunistic pathogen *Legionella* is important to assess the risk of infection and plan remediation actions. While monitoring is traditionally carried on through cultivation, there is an ever-increasing demand for rapid and high-throughput molecular-based approaches for *Legionella* detection. This paper provides valuable insights on how DNA extraction affects downstream molecular analysis such as the quantification of *Legionella* through droplet digital PCR and the characterization of natural microbial communities through sequencing analysis. We analyze the results from a risk-assessment, legislative, and ecological perspective, showing how initial DNA processing is an important step to take into account when shifting to molecular-based routine monitoring and discuss the central role of consistent and detailed reporting of the methods used.

**KEYWORDS** *Legionella*, DNA extraction, monitoring, pathogen quantification, ddPCR, sequencing

Accurate pathogen detection and quantification in drinking water is necessary to establish effective policies and minimize the infection risk. *Legionella* is a large genus of waterborne bacteria comprising more than 60 species, of which approximately half have been identified as opportunistic human pathogens (1). The incidence of Legionnaire's disease has been increasing worldwide in the past decades. In Switzerland,

Address correspondence to Frederik Hammes, frederik.hammes@eawag.ch.

The authors declare no conflict of interest.

See the funding table on p. 14.

the reported incidence rate per 100,000 population was 7.24 in 2023, compared to 6.61 in 2017 and 3.54 in 2014 (2). Current monitoring strategies heavily rely on culture-based methods (3), which have well-documented limitations, including being time consuming, producing less precise results, and having a bias toward *Legionella pneumophila* (4). Molecular-based methods [i.e., quantitative/droplet digital PCR (qPCR/ddPCR)] offer an alternative to culture that addresses some limitations associated with culture-based methods but have inherent limitations of their own. For example, molecular methods can be used to obtain more precise results faster, but detect genetic material from both live and dead organisms. While some legislation already allows for molecular methods to be used for compliance with monitoring requirements (5), the routine use of such methods remains a future prospect.

DNA extraction represents an essential step for molecular quantification of organisms that affects every downstream molecular analyses. DNA extraction methods consist of (i) lysis of cell membranes and nucleus to release DNA, (ii) separation of DNA from other cellular components and debris, and (iii) purification of the DNA. How well these steps are executed affects DNA yield, quantification of target organisms, and characterization of microbial community composition from various matrices (6–9). Although all the DNA extraction methods share these general procedures, numerous commercial kits, protocols, and *ad-hoc* adaptations are available and described in literature. For example, protocols.io, a large repository of laboratory protocols contributed by researchers, has more than 2,000 entries for DNA extraction methods and optimizations (10). While different commercial kits and adaptations are usually implemented to extract DNA from specific matrices, thus not performing optimally on others, it is notable that there are examples where even different versions of the same commercial kit can produce different amounts and qualities of extracted DNA (11). Despite the documented differences introduced by the DNA extraction on downstream analysis and results, no information on how different DNA extraction procedures influence the quantification of *Legionella* spp. is available to our knowledge.

There is generally a lack of adequate reporting of DNA extraction recoveries and potential biases in the drinking water field, as highlighted by the Environmental Microbiology Minimum Information (EMMI) guidelines (12). To address this, the EMMI guidelines propose the use of negative and positive controls to identify potential contaminants or issues with the DNA extraction process. An external positive control is particularly important to quantify extraction efficacy. The guidelines appropriately leave the specific choice of control to the study authors (12), but as a result, there is no established practice for *Legionella* DNA extraction from environmental samples. Using a pure culture, synthetic mixture of pure cultures, or cultures phagocytized by a host organism are all logical and common choices. However, these would provide little information about how well the extraction method extracts *Legionella* DNA from an environmental sample with a complex mixed microbial community or about potential biases it introduces when performing sequencing analysis. From an ecological point of view, little information about the influence of DNA extraction on the microbial community characterization in drinking water systems is available, particularly when wanting to distinguish between water and biofilm phases.

In this study, we processed water and biofilm samples collected via a community science sampling campaign using two DNA extraction methods common in drinking water studies to demonstrate how the variability between and among methods can impact environmental sampling interpretation, with a specific focus on the quantification of *Legionella* spp. and community structure.

## MATERIALS AND METHODS

### Field-scale community science sample collection and processing

#### *Participant recruitment and sampling*

Employees from two research institutes were recruited via Listserv email to collect shower water and biofilm samples from their homes. Participants received a pre-labeled sampling kit that contained a sterile 1-L glass bottle, a new shower hose to replace their existing one, sealing caps to retain the water within their harvested shower hose, and detailed sampling instructions. Briefly, following >8 h of stagnation, the participant opened the shower tap and filled the 1-L bottle with the first flush of hot water until the bottle was full but not overflowing. The participant then removed the shower head and detached the existing shower hose, keeping the ends at approximately the same height to prevent water in the hose from leaking out. The used shower hose was capped with new threaded PVC caps to retain the water inside the hose and replaced with a new one. Samples were delivered to the lab for processing within 24 h. The experimental workflow is shown in Fig. 1.

#### *Sample processing—water*

After mixing the 1-L sample, 100 mL was aliquoted to a sterile glass container, total cells were quantified using flow cytometry, and *L. pneumophila* was quantified using IDEXX Legiolert liquid culture kit (IDEXX Laboratories, Inc, Westbrook ME, USA) according to the manufacturer's instructions. The remains of the water sample were filter-concentrated onto duplicate 0.2-µm polycarbonate filters (Steriltech Corporation, Auburn MA, USA),

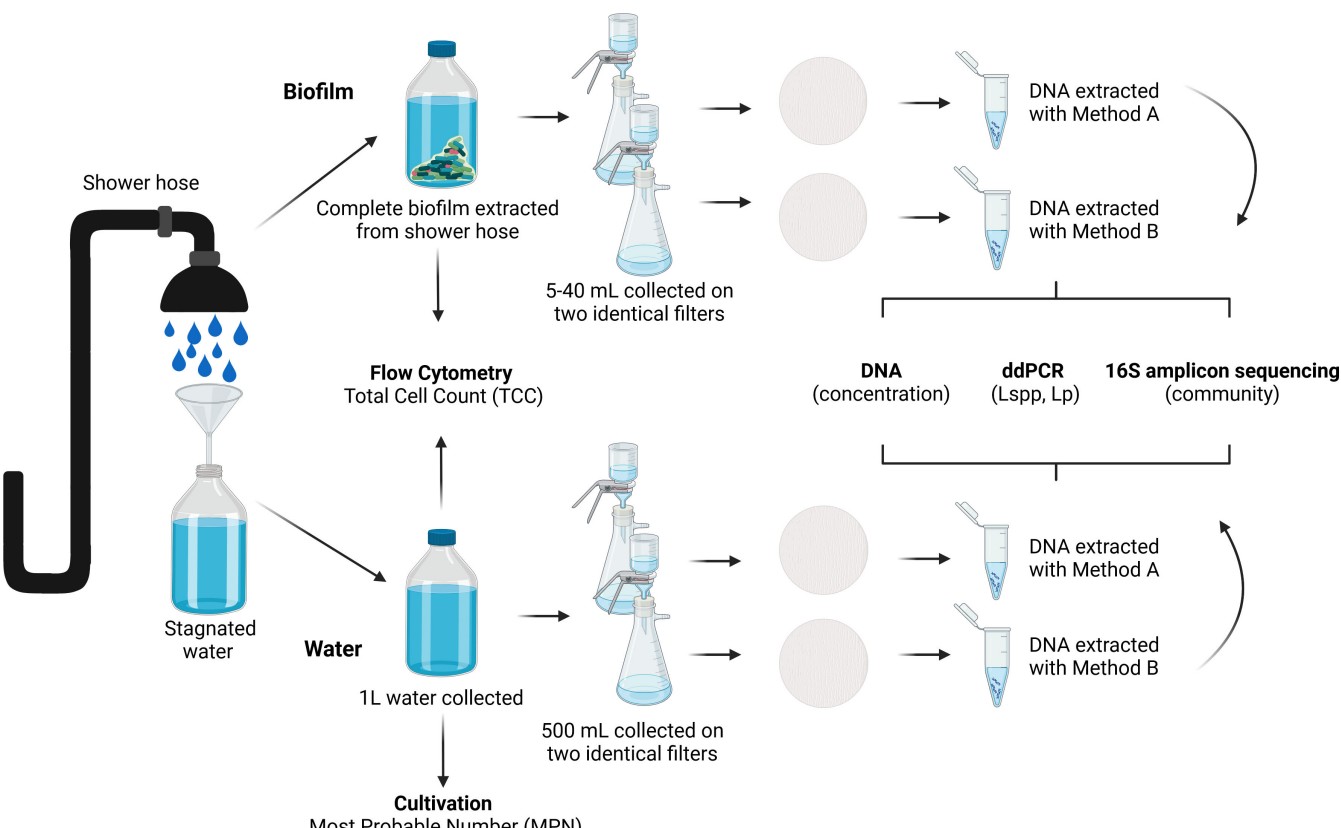

**FIG 1** Overview of the experimental workflow used in this study. The shower hoses were collected with the first liter of 8-h stagnated water. The biofilm was extracted from the shower hoses, and then, the biofilm and the water phase were (i) used for flow cytometry to measure the total cell count (TCC) and (ii) filtered twice onto two different filters from which the DNA was extracted using two different commercially available kits (Image: biorender.com).

recording the volume filtered; each filter was fragmented using a flame-sterilized scalpel and tweezers, and frozen at −20°C until DNA extraction was carried out.

### Sample processing—biofilms

Water contained in the shower hose was collected, and the hose exterior sheath (if applicable) was removed. The hose was filled with 20 mL of autoclaved 2-mm glass beads, and biofilm was eluted into filter-sterilized and autoclaved water through five rounds filling the hose with the sterile water, with sonication of the hose using the vibration of the beads to suspend the attached biomass, and then collecting the water within the hose after each sonication. Each time the water was collected, the end from which the water flowed from the hose was alternated to create as much flow reversal and mixing as possible. After the last round of sonication/collection, the glass beads were removed, the hose filled with 50 mL of water a final time, inverted 30 times, and water was collected into the sample bottle. The total amount of water used for biofilm elution, and the length and diameter of the hose were recorded. Total cells were measured in the suspended biofilm sample after three rounds of 30 s of sonication of a 2-mL aliquot at 40% amplitude and diluting 1:1,000 or 1:10,000 as necessary. Eluted biofilm from the sample bottle was filter-concentrated onto duplicate 0.2-µm polycarbonate filters until clogging (5–70 mL), each filter fragmented using a flame-sterilized scalpel, and frozen at −20°C until DNA extraction was carried out.

### DNA extraction

Two commercially available DNA extraction kits (hereafter indicated as Methods A and B) with the same adaptation were used on each water and biofilm sample set. Each kit is commonly used for extracting DNA from environmental water samples. Both Methods A and B are spin-column-based DNA extraction procedures, consisting of similar protocols, which involve the use of a chemical lysis solution, a beat-beating step and a binding matrix solution prior to spin-column concentration, washing via centrifugation, and a final elution in 100 µL. The specific composition of the reagents provided for each steps was not disclosed by the respective manufacturers. Adaptations to the manufacturer instructions to each kit included submerging fragmented filters in 6 µL of Lysozyme (50 mg/µL; Thermo Fischer Scientific, Waltham MA, USA) and 294 µL of 1× TE buffer for 1 h at 37°C mixing at 300 rpm, adding 30 µL of Proteinase K (20 mg/mL; Thermo Fischer Scientific, Waltham MA, USA) and 300 µL of DNA extraction kit cell lysis solution, and continuing incubation for 30 min at 56°C mixing at 300 rpm, and adding 600 µL of chloroform (isoamyl alcohol, 24:1 suitable for nucleic acid purification) with DNA extraction kit lysis beads. These adaptations were previously described by Voslo et al. (13). The beads, dissolved filters, and final 1,230 µL of solution were then bead beaten on a vortex shaker at maximum speed for 5 min. Afterward, manufacturer instructions were followed. Each time DNA extraction was performed, a DNA extraction negative (un-used filter) and positive control was processed. The positive control consisted of replicate 100 mL bulk water samples collected from a bioreactor colonized by native *Legionella* (additional details provided in Fig. S5). The bioreactor water was collected in bulk, then 100-mL aliquots were concentrated onto replicate filters and frozen until DNA was extracted. *L. pneumophila* liquid culture and total cells were quantified in the bioreactor water to provide independent comparison of the extraction efficiency of *L. pneumophila* and total cells. The impact of each pre-treatment step on the DNA extraction has been evaluated and reported in Fig. S1.

## Water quality measurements and methodology

### Total cells counts and extracted DNA

Total cells were quantified using a CytoFLEX (Beckman Coulter, Inc., Brea CA, USA) flow cytometer in 250-µL aliquots stained using SYBR® Green I (SG, Invitrogen AG, Basel, Switzerland; 10,000× diluted in Tris buffer, pH 8). Stained cells were incubated for 15 min

at 37°C prior to analysis (14). Extracted DNA was measured using a Qubit dsDNA HS assay (Thermo Fischer Scientific, Waltham MA, USA), with a linear detection range of 0.2–100 ng double stranded DNA.

### *Legionella* spp. and *L. pneumophila* gene copy enumeration

*Legionella* spp. (ssrA) and *L. pneumophila* (mip) were measured with (ddPCR) using gene targets based on previously published assays validated to ISO SO TS12869:201 (15, 16) and adapted for the ddPCR platform (17). Primer and probe sequences, master mix composition, and thermocycling conditions can be found in the supplemental material (Table S1). A ddPCR reaction negative control (DNAse-free water) was included for each batch of master mix prepared and was always negative. A ddPCR reaction positive control (Centre National de Référence des Légionelles) was included on each thermo-cycling run. Droplet formation and PCR thermocycling were performed using a Stilla geode (Stilla Technologies, Villejuif, France) and read using a Prism6 analyzer with Crystal Reader software imaging settings pre-set and optimized for PerfeCTa multiplex master mix (QuantaBio, Beverly MA, USA). Droplets were analyzed using Crystal Miner software. Only wells with enough total and analyzable droplets, as well as a limited number of saturated signals, were accepted according to Crystal Miner software quality control. Positive droplets were delineated using polygons, with positive wells being considered as those resulting in at least three droplets within the polygon. The limit of detection was 5 gc/reaction (1 gc/µL of template), and the limit of quantification was 12 gc/reaction (2.4 gc/µL of template 17). Any sample with significant intermediate fluorescence clusters (i.e., "rain") was diluted 1:10 and rerun.

### 16S rRNA amplicon sequencing

For sequencing, the V4 region of the 16S rRNA gene was amplified by PCR using the primers Bakt_515F–Bakt_805R (18), and the DNA was quantified by Qubit dsDNA HS Assay (Thermo Fischer Scientific, Waltham MA, USA). Samples were diluted, where possible, to the concentration of 1 ng/µL. A two-step PCR protocol was used to prepare the sequencing library: a first amplification (target PCR) was carried out with 1× KAPA HiFi HotStart DNA polymerase (Roche, Basel, Switzerland), 0.3 µM of each 16S primer, and 2 µL of template DNA. After amplification, the PCR products were purified with the Agencourt AMPure System (Beckman Coulter, Inc., Brea, USA). The second PCR (adaptor PCR) was performed with limited cycles to attach specific sequencing Nextera v2 Index adapter (Illumina, Inc., San Diego CA, USA). After purification, the products were quantified and checked for correct length (bp) with the High Sensitivity D1000 ScreenTape system (Agilent 2200 TapeStation; Agilent Technologies, Inc., Santa Clara CA, USA). Sample concentration was adjusted, and samples were subsequently pooled together in a library at a concentration of 4 nM. The Illumina MiSeq platform was used for pair-end 600 cycles (16S) with 10% PhiX (internal standard) in the sequencing run. Negative controls (PCR-grade water) and a positive control (self-made MOCK community) were incorporated. Primer sequences, master mix composition, and reaction conditions can be found in the Supplementary Information (Table S2). These steps were performed in collaboration with the Genetic Diversity Centre (GDC) of ETH Zurich.

### Data analysis

16S rRNA sequencing data were processed on HPC Euler (ETHZ) using workflows established by the GDC (ETHZ, Zurich). Detailed data-processing workflows are provided in the supplementary materials. For the 16S data set, all R1 reads were trimmed (based on the error estimates) by 25 nt, the primer region removed, and quality filtered. Ultimately, sequences were denoised with error correction and chimera removal and amplicon sequence variants established using UNOISE3 (19). In this study, the predicted biological sequences will be referred to as zero-radius operational taxonomic units (zOTUs). Taxonomic assignment was performed using the Silva 16S database (v128) in

combination with the SINTAX classifier. Samples were not rarefied to avoid the loss of data due to differences in the sequencing depth (20). Distance ordination and relative abundance were calculated using R (version 4.2.1) and R studio (version 2022.07.2 + 576) using the Bioconductor package "phyloseq" (version 1.42.0), (21). Linear discriminant analysis effect size (LefSe) analysis was performed using Microbiome Analyst (22). SparCC correlation analysis was performed using the software FastSpar (23). All graphs were constructed with the R package "ggplot2" (version 3.4.0). Unless otherwise specified, all packages were operated using the default settings. Absolute abundance for *Legionella* quantification was calculated as follows:

Absolute Abundance = Relative Abundance (16S amplicon sequencing) × Total Cell Count (Flow Cytometry) × DNA extraction efficacy

## RESULTS

### Comparison of DNA extraction efficacies

The two DNA extraction methods used in this study yielded different amounts of DNA (Fig. 2). Method A extracted substantially more DNA overall (median value 1.03 fg/cell, IQR: 1.70; *n* = 83) than method B (median value 0.03 fg/cell, IQR: 0.06; *n* = 74) (Wilcox paired test—*P*-value: <0.001). The median DNA/cell value detected with method A was 186-fold higher than that of method B for water samples, but only 13-fold higher for biofilm samples. However, this could partially be artificial due to the fact that biofilm samples seem to reach a plateau at approximately $10^4$ ng of extracted DNA (see Water and biofilm phase). Method B failed to extract detectable levels of DNA from 15 out of 50 water samples. These samples with low/no extracted DNA were still used for *Legionella* quantification through ddPCR, but were excluded from the sequencing run.

To calculate the extraction efficacy for each sample, an average DNA-per-cell value of 4 fg/cell was used (24). The median value of the extraction efficacy for the water samples

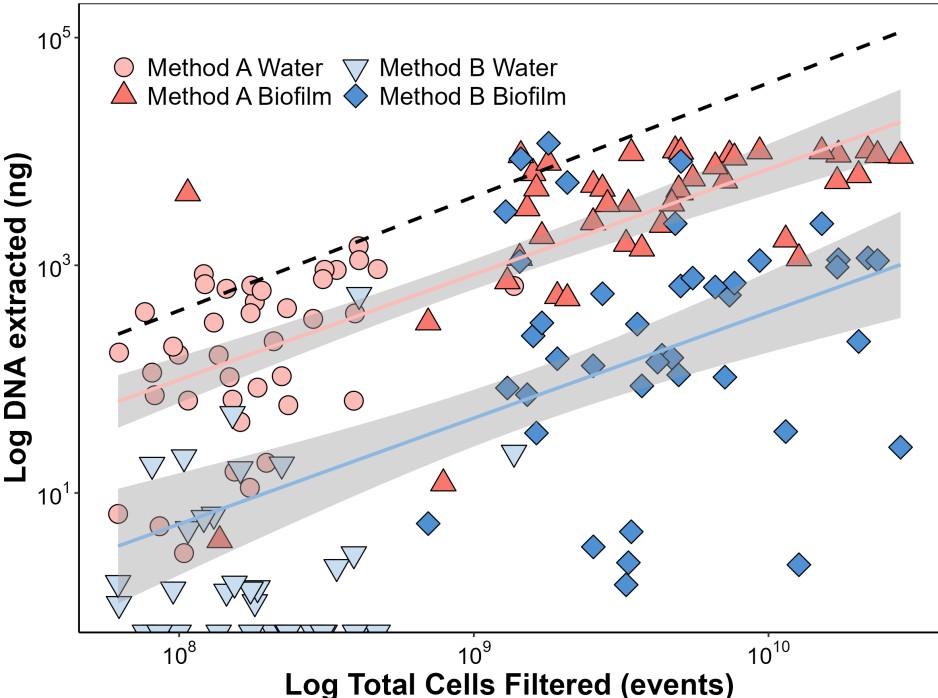

**FIG 2** DNA extraction efficacy of two tested extraction methods on the same water and biofilm samples collected from shower hoses in private homes. Method A samples are red, while method B samples are blue. The samples shown in lighter color refer to the water phase, while the darker color indicates the biofilm phase. The dashed line represents a theoretical maximum DNA extraction efficacy of 4 fg/cell. Markers crossing the axes represent samples with no quantifiable extracted DNA.

extracted with method A was 35%, while it was 0.2% for method B. For biofilm samples, the median value was 21.7% for the ones extracted with method A and 1.6% for method B. A few samples had an extraction efficacy above 100%, which is likely because the chosen value of 4 fg/cell does not consider the heterogeneity in DNA content across different bacteria, particularly in complex communities (25).

## Impact on *Legionella* quantification

The differences in the DNA extraction affected the detection and quantification of *Legionella* spp. and *L. pneumophila*. The quantification of *Legionella* spp. using ddPCR was significantly higher (Wilcox paired test—*P*-value < 0.001) for the samples extracted with method A than the ones extracted with method B, showing a 39-fold median increase of gene copies detected per 100 mL (Fig. 3a). *Legionella* spp. was not detected with ddPCR in 21 water samples from method B and 10 samples (one water and nine biofilm) from method A (Fig. 3a), but in all cases where *Legionella* spp. was not detected, it was detected in the same samples with the other method. In nine samples among the 21 non-detected by method B, there was no quantifiable DNA in the first place (Comparison of DNA extraction efficacies). As with *Legionella* spp., the quantification of *L. pneumophila* using ddPCR was significantly higher (Wilcox paired test—*P*-value < 0.001) for the samples extracted with method A than the ones extracted with method B, with a 44-fold increase of gene copies per 100 mL detected (Fig. S2).

Overall, there was a 50% and 45% of presence–absence agreement between ddPCR and cultivation in the samples for methods A and B, respectively (Fig. S3). As expected, cases were observed, in which both methods detected *L. pneumophila* DNA, while the cultivation data resulted negative. Method A had 16 samples that were positive by ddPCR but negative by culture (*n* = 43, 37%); method B had 10 (*n* = 42, 24%). However, there were also samples positive by culture but not by ddPCR. Method A had six samples that were culture positive but ddPCR negative (13%), while method B had 13 (31%). While this could be attributed to low DNA extraction efficacy for the 13 samples extracted with method B, the six samples extracted with method A showed overall good recovery (DNA extraction efficacy >50%), which cannot justify this observation.

When determining the relative abundance of *Legionella* spp. (Fig. 3b) from sequencing data, highly variable results were observed between methods. *Legionella* spp. was not detected through amplicon sequencing in six samples (four water, two biofilm) extracted with method B, and in four samples (two water, two biofilm) extracted with method A, and one sample with both methods. For a direct quantitative comparison with the ddPCR data, absolute abundance was calculated by multiplying by the TCC while

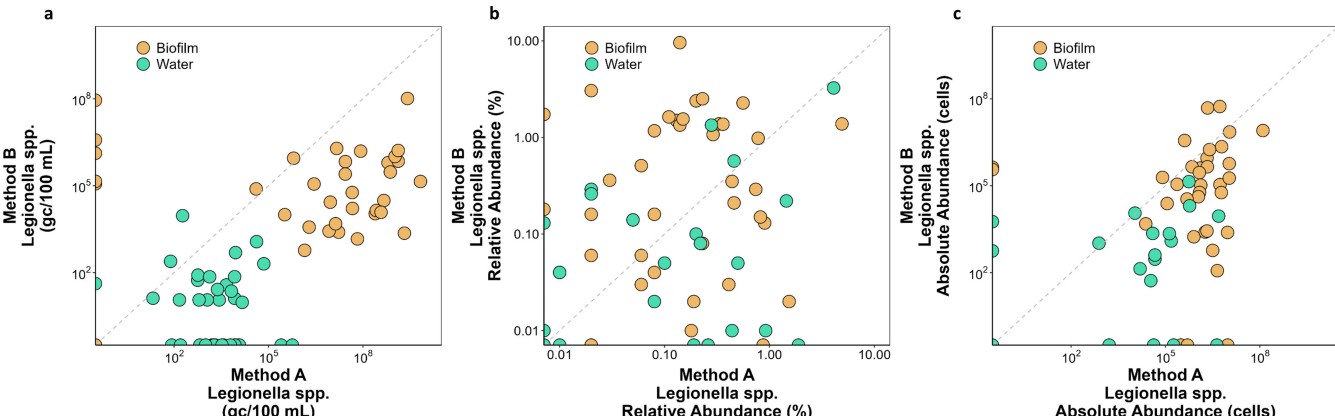

**FIG 3** Comparison of two DNA extraction methods for the quantification of *Legionella* spp. (a) Number of gene copies per 100 mL using ddPCR. (b) Relative abundance of *Legionella* spp. calculated with 16S amplicon sequencing data. (c) Absolute abundance of *Legionella* spp. calculated by multiplying the relative abundance (shown in b) with the total number of filtered cells and the DNA extraction efficacy (Fig. 2). Biofilm samples are indicated in orange, while water samples are shown in green. Markers crossing the axes represent samples with no detected *Legionella* spp.

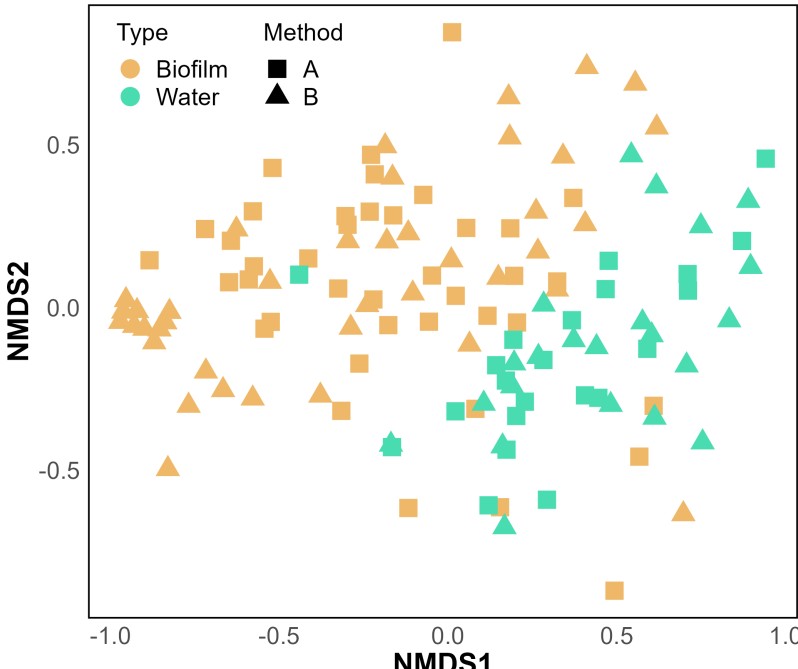

**FIG 4** Ordination analysis of the 16S amplicon sequencing data, stress = 0.222. The graph shows distance among samples calculated using the Bray–Curtis method and plotted on an NMDS plot. The color of the points refers to the phase (biofilm in orange; water in green), while the shape shows the method used to extract the DNA of the samples (Method A indicated with squares; method B indicated with triangles).

accounting for the DNA extraction efficacy (Fig. 3c). Quantification of *Legionella* spp. using amplicon-sequencing-derived absolute abundance remained statistically higher for method A than for method B, showing a 50-fold difference (Wilcox paired test— *P*-value: <0.001).

## Impact on ecological observations

Besides *Legionella* quantification, the differences in the extraction methods also affected the community structure detected in the samples: bacterial genera were detected with different abundances in samples extracted with either method A or B. The distance between samples was calculated using the Bray–Curtis index, which resulted in an Non-metric Multi-Dimensional Scaling (NMDS) plot with a stress value of 0.222 (Fig. 4). The analysis showed more significant clustering according to the phase of the samples [i.e., water or biofilm; permutational multivariate analysis of variance (PERMANOVA), *P*-value < 0.001, *R*2 = 0.03], though the samples also cluster significantly with respect to the two DNA extraction methods used (PERMANOVA, *P*-value < 0.001, *R*2 = 0.02). This suggests that the selection of extract kit changes the community composition.

To provide a better overview of the ecological differences caused by the different DNA extraction methods, LefSe analysis was used to determine the genera that are most likely to explain differences between the two (Fig. 5) (26). The analysis shows that the relative abundance of 52 genera are enriched in either method A (32 genera) or method B (20 genera) with linear discriminant analysis (LDA) scores ranging between 2 and 5. Based on the effect size, we detected the genera *Sphingomonas*, *Thermus*, and *Gemmata* to be the most enriched in method A (LDA scores: 5, 4.81, 4.59), while *Caulobacter*, *Obscuribacteraceae*, and *Hirschia* were found to be the genera with a stronger effect size for method B (LDA scores: 5.05, 4.87, 4.52) (Fig. 5a).

While the genus *Legionella* was not detected by this statistical analysis as enriched in one specific method (Fig. 5g), a LefSe analysis performed at the zOTU level was able to detect six *Legionella*-associated zOTUs (zOTU492, zOTU620, zOTU1267, zOTU1386,

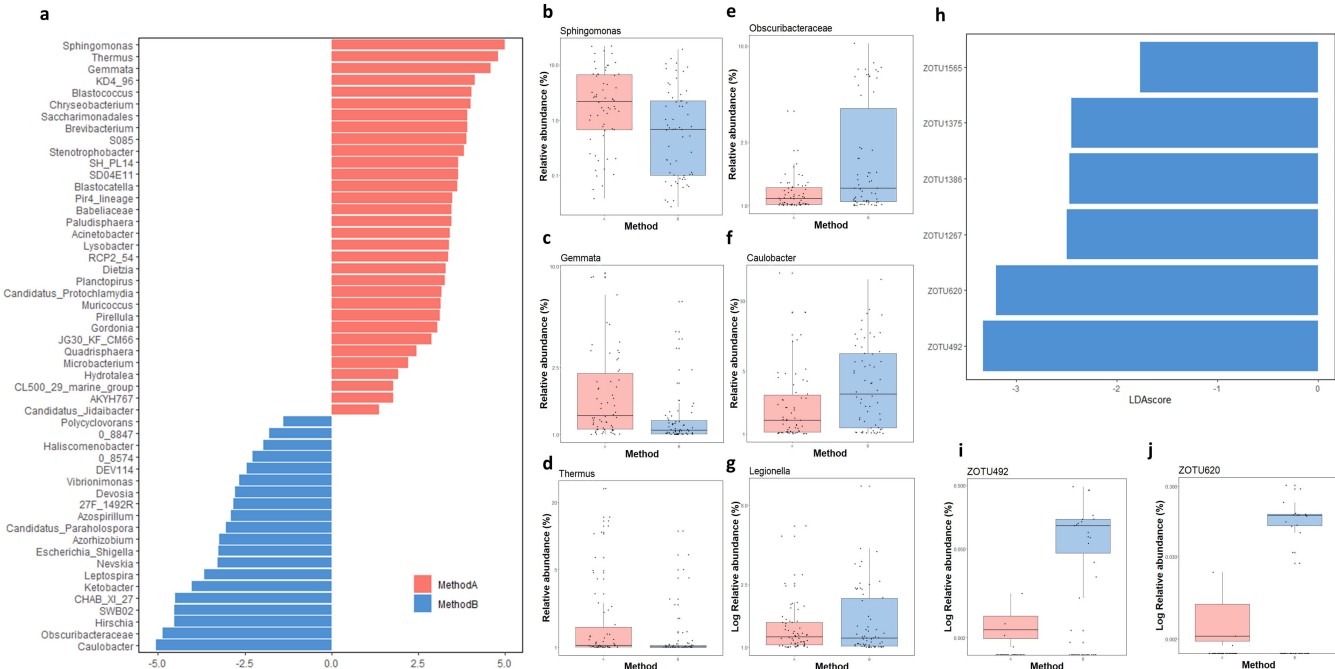

**FIG 5** Differential abundance of bacterial genera and *Legionella*-associated zOTUs between methods A (in red) and B (in blue). (a) LDA scores derived from the LefSe analysis showing the genera that are overall enriched in one method. (b–d) Box plots of the relative abundance obtained using the two methods of the genera with the highest absolute LDA scores in Fig. 5a; (g) Box plot for the relative abundance of *Legionella*. (h) LefSe analysis showing the LDA scores of *Legionella*-associated zOTUs that are enriched in either methods. (i, j) Box plot of the relative abundances of the two *Legionella*-associated-zOTUs with the highest absolute LDA scores in Fig. 5h.

zOTU1375, zOTU1565) whose relative abundances were all enriched in method B (LDA score ranging 1.77 to 3.33), but not in method A. In one case, a *Legionella* zOTU (zOTU1267) was classified to the species level, and assigned to *L. geestiana*, while the other five zOTUs only had a classification to the genus level. SparCC analysis was performed to detect correlations between the zOTUs linked to the genus *Legionella* and the ones representing the rest of the community. For the biofilm samples extracted with method A, 53 significant correlations (correlation coefficient >0.4) were detected ($n$ samples = 45), while 3,442 correlations were detected in the samples extracted with method B ($n$ = 45). For the water phase, 83 significant correlations were detected in the samples extracted with method A ($n$ = 39), while 225 were detected in the ones extracted with method B ($n$ = 22). Among the correlations detected, five were shared between the methods in the biofilm phase, while 35 were shared in the water samples.

## DISCUSSION

### The choice of the DNA extraction method is important

DNA extraction is a crucial step for downstream molecular analysis and has been identified as one of the most problematic steps, since it is prone to introduce biases and inefficiencies in extracting DNA (27, 28). Ineffective DNA extraction with lower DNA yields (Fig. 2) results in lower ddPCR concentrations for specific target organisms (Fig. 3a). This was previously demonstrated by Cerca and colleagues (6) showing that DNA extraction efficacy is likely to influence the quantification of target organisms by qPCR in polymicrobial consortia. The authors extracted gDNA from three microorganisms in pure cultures at known concentrations (individually or collectively cultured) and performed qPCR generating different calibration curves, of which only the one that corrected for the DNA loss during the DNA extraction provided reliable results.

Several previous ecology studies noted the potential influence of DNA extraction methods on DNA yield, bacterial diversity, and relative abundance measurements, for example, in soil samples (7), human breast milk (8), and faecal samples (9). However, only a few studies have so far investigated the impacts of different DNA extraction methods on drinking water samples (28, 29). While each of these studies identified the method that fits the need of the proposed experimental design, a perfect universal DNA extraction method does not exist at present, and the efficacy of the extraction protocol applied is dependent on the sample type or presence of specific organisms (27, 30, 31). While these methodological differences are not surprising *per se*, little data are available on how DNA extraction influences detection and quantification of microorganisms in drinking water, how this specifically affects the quantification of the opportunistic pathogenic *Legionella* species, and how these challenges/problems can be mitigated if molecular methods are to be used quantitatively for routine monitoring within legislative frameworks.

We found that different methods extracted significantly different amounts of DNA from the same samples (Fig. 2) and that this consequently influenced the quantification of *Legionella* via ddPCR (Fig. 3a). We furthermore showed the extraction efficacy relative to an ideal situation in which all cellular DNA is extracted using an average estimation of cellular DNA content (Fig. 2) (24). However, this is an approximation that does not account for the variations in DNA content among different bacteria and requires access to a flow cytometry to obtain the total cell count. Other studies report the addition of external spike-in sequences to measure recovery of a known amount of target (32). This approach does not provide, however, any information about the correct lysis of the cells. The latter can be achieved by spiking whole-cell microorganisms at a defined amount, which provides quantifiable recovery of a target organism (or surrogate microbes) and is an approach also advocated for in the EMMI guidelines (12). Regardless of the approach used, here, we advocate for consistent measurement and reporting of the extraction efficacy and the way it has been calculated, particularly in studies where quantitative data are generated. Such information allows an estimation of the DNA loss and the potential for a corrective factor to be applied to quantitative data.

## Water and biofilm phase

The water and biofilm phases revealed interesting differences in terms of DNA extraction. Both methods reported a lower DNA yield in DNA-per-cell for the water samples compared to the paired biofilm samples, but this effect was more pronounced for method B, which failed to extract detectable DNA from multiple water samples (Fig. 2). While the lower yield can be partially attributed to the presence of extracellular DNA in the biofilm matrix (33), the poor extraction efficacy recorded overall with method B can only be attributed to the weak performance of the method itself. We observed a maximum DNA recovery of approximately $10^4$ ng, which appears to be an artificial limit at high cell concentrations (Fig. 2). One possible explanation for this could be that there is a maximum amount of nucleic acid that can bind to the extraction column used, which is referred to as binding capacity. While every kit has a distinct binding capacity, the plateau observed in our results is consistent to that reported by several manufacturers, and this can potentially lead to an underestimation of DNA concentrations and, by consequence, of specific target organisms when processing high biomass samples.

Our study also revealed a clear separation in the microbial communities detected in the water and in the biofilm phase, irrespective of the extraction method that was used (Fig. 4). This is an important ecological observation, and only a few other papers have previously reported that different communities inhabit the two phases in a distinctive way. In an ecological investigation of the communities in an unchlorinated distributions system, Liu and colleagues (34) showed that bulk water and pipe biofilm clustered differently in a principal component ordination analysis, identifying key species dominating either the water (e.g., *Polaromonas* spp.) or the biofilm (e.g., *Pseudomonas* spp., *Sphingomonas* spp.). Similarly, Proctor and colleagues observed that the biofilm

and bulk water bacterial communities were significantly different in shower hoses (35). In particular, it was shown that three taxa accounting for 91% of the biofilm sequences only accounted for 31% of the cold water sequences detected, while seven predominant taxa in the cold water accounted only for 2% of the biofilm sequences detected. One possible explanation for this phenomenon can be attributed to early establishment of defined ecological niches within the biofilm compared to higher dynamics of bulk water due to flow and exchange of nutrients. Biofilms have lower richness than the bulk water, meaning that only a few species dominate the biofilm ecological niches (36). This suggests that there is relatively low ability for transient bulk water organisms to establish in developed biofilms. This could also explain why it has been observed that the communities of biofilms are more stable in time than the ones present in water (37).

With respect to *Legionella*, these differences are important as it is known that the pathogen lives at higher concentrations in biofilms (38, 39), as is the case with most opportunistic building plumbing pathogens (40). We observed this phase difference despite the fact that all shower hoses in our experiments were collected after stagnation, thus perhaps providing more opportunities for an exchange between the two phases. However, the result could partially be attributed to the sampling strategy adopted: we collected 1 L of water samples, thus not only the water that was stagnating in the shower hose (approximately 80 mL). The additional water coming from building plumbing, apart from the shower hoses, might have masked any exchange between the two phases to some extent, thus increasing the magnitude of the separation observed (Fig. 4). This highlights the importance of understanding and correctly reporting all upstream processing steps that may influence downstream analysis; it also demonstrates the importance of how considering both phases in ecological studies and routine monitoring can provide more information than typical monitoring plans. Previous studies have already adopted strategies for on-site removal and replacement of portions of biofilm-containing pipes, through dedicated points of entrance and aseptic insertion of new coupons (41). However, it is not possible to establish how small portions can be representative of the entire distribution network biofilm, which is an issue that must be further investigated.

## Relative and absolute abundance

To quantitatively compare the abundances of microorganisms across different samples and studies analyzed with 16S amplicon sequencing, it is useful to convert the relative abundances values into absolute abundance (42). Several studies have reported the potential occurrence of opportunistic pathogens in environmental samples using sequencing data in the form of relative abundance (39, 43, 44). While these observations are ecologically relevant, the use of relative abundance does not provide quantitative information on concentrations. An estimation of the absolute abundance can be obtained using different methods, of which the most used are (1) qPCR quantification of the total 16S rRNA gene and (2) flow cytometric total cell counts, chosen for this study (45, 46). Some considerations are, however, necessary. Since both qPCR and sequencing use the same extract that is subjected to the same methodological biases, calculating the absolute abundance with 16S qPCR data has the downside of carrying any DNA loss occurred during the extraction, leading to the generation of inaccurate results. In this case, it is therefore necessary to account for the DNA loss during the extraction (i.e., determining the DNA extraction efficacy) for the calculation of the absolute abundance. Moreover, calculating the absolute abundance with qPCR harbors additional limitations: previous studies have, in fact, observed that qPCR can detect only large changes in gene concentrations and is strongly influenced by the primer pairs and the reaction conditions (42). The quantification using flow cytometry, on the contrary, can overcome these limitations, as it is not necessary to calculate the DNA extraction efficacy separately to obtain the absolute abundance.

## Relevance of an accurate *Legionella* detection and quantification

The use of different DNA extraction methods clearly has variable outcomes in relation to the accurate detection and quantification of target organisms (in this case *Legionella* spp.) in drinking water systems. This is relevant from two perspectives: (i) the implications for microbial ecology studies and (ii) the implications for the monitoring of *Legionella* spp. under regulatory settings.

### Ecology

Ecological studies often work toward providing insights into the microbial composition of given samples under specific conditions, but an accurate detection of the organisms involved and their relative proportions are crucial to define valuable biological interpretations of the data collected. The most common reported examples of how DNA extraction affects ecological observations are related to the studies conducted on the human microbiome. For example, in a study aiming at establishing the impact of DNA extraction procedures on the assessment of human gut composition, Kennedy and colleagues demonstrated that, within individual patients, community structures clustered together based on the kit used to extract the DNA from the samples (47). This effect was even more pronounced depending on the distance metrics that were used, and this had an important value given that the study was aimed at showing differences in community composition between volunteers and patients with inflammatory bowel disease (47). In a different study involving the human oral microbiome, Lazarevic and colleagues found that while the most abundant taxa were detected in the samples extracted with both methods, for some genera, the relative abundances were significantly different depending on the kit used (48). Similar results were obtained in a study that compared DNA extraction kits and primer sets for freshwater sediments samples: no significant diversity in terms of community structure was, in fact, detected, but the relative abundance of specific taxa varied significantly (49). Moreover, this study also highlighted differences in richness and relative abundance for the eukaryotic communities detected in samples extracted with two different kits.

In the context of the microbial ecology of *Legionella*, the relevance of this is mainly linked to the observational studies aiming at understanding how this opportunistic pathogen lives in aquatic systems through their relationship with the surrounding organisms (44, 50–52). In a previous paper from our group, for example, we observed that *Legionella* spp. correlated positively and negatively with prokaryotic and eukaryotic microorganisms in biofilm samples from plumbing systems (39). These studies used different combinations of statistical approaches that process the sequencing outcome (often in the form of relative abundance) to infer correlations. In this context, biases introduced by the DNA extraction (in terms of community structure, abundance and amplicon sequence variants detected) can influence the analysis leading to wrong ecological interpretations. Our results, for example, demonstrate that the correlations inferred with SparCC when using the sequencing data of samples extracted with different methods produce variable outcomes, both in the number of correlations and identity of the zOTUs involved. We argue that a correct understanding of the microbial ecology is crucial for the control of the pathogen in drinking water systems, and the possibility of comparing studies is an important tool toward this goal; however, biases due to the molecular protocol applied (i.e., DNA extraction) can work against the formulation of correct ecological hypothesis.

### Legislative compliance and risk assessment

A reliable quantification of *Legionella* spp. is important to accurately control the level of the pathogen in engineered aquatic systems and to assess the risk linked to its presence. Quantitative microbial risk assessment (QMRA) uses information regarding pathogen concentration to determine health implications of microbial hazards (53). Thus, the concentrations of the pathogenic organism investigated are of extreme importance to

establish the risk associated with its exposure. With respect to QMRA of *Legionella*, most studies use concentrations measured with conventional culture approaches, which likely underestimates actual concentrations and, in turn, may lead to the underestimation of risk (53). Molecular methods (including qPCR or ddPCR) are generally more sensitive and overcome some of the limitations of traditional culture approaches, but, as demonstrated in this study, are subject to errors arising from DNA extraction efficacy differences. Therefore, the future inclusion of molecular methods in QMRA requires careful consideration, documentation, and reporting of the entire sampling processing pipeline.

Similarly, the effects of the DNA extraction on the quantification of *Legionella* can also have an impact on the routine monitoring for the presence of *Legionella*. Normally, the water phase is collected, filtered, and then plated onto buffered charcoal yeast extract (BCYE) agar plates for enumeration and culture confirmation (3). Investigations of the presence of *Legionella* in environmental samples, which can be used as a surveillance strategy, are often carried out using a culture-dependent approach. For example, several studies have reported environmental monitoring of *Legionella* in hospital settings over multiple years, to prevent nosocomial infections and link cases of Legionnaires' disease with the environmental source of infection (54–56). Interventions are required when the concentration of *Legionella* reaches the threshold indicated by the national authorities (in Switzerland, 1,000 colony-forming units (CFU)/L (57). This entire process, however, is time and labor consuming as culturing *Legionella* typically takes 7–14 days. Moreover, culturing does not account for bacteria in viable-but-not-cultivable state (58). Therefore, the demand for the implementation of molecular methods (i.e., quantification of *Legionella* through qPCR/ddPCR) in the context of assessing water quality has increasingly spread across practitioners and authorities, and in fact, for example, the new EU legislation allows for alternative methods to be used (5). The implementation of such molecular methods in routine monitoring calls for a more detailed and standardized reporting of the protocols used.

Our data not only demonstrates the variability in terms of concentration of *Legionella* when using different DNA extraction methods but also highlights how the pathogen is not detected in some samples extracted with one method, while it is quantified when the DNA is extracted with the other method used in this study. This would obviously influence the legal settings, since a wrong estimation of the concentration of *Legionella* due to extraction biases can lead to either unnecessary interventions (which are expensive for the practitioners) or increased risks. While being aware of the challenges involved in the standardization of methods, we strongly believe that an extensive reporting of the protocol details (e.g., DNA extraction method used, extraction efficacy, and how it was calculated) would be beneficial for a reliable comparison among studies, monitoring strategies and regulations.

## ACKNOWLEDGMENTS

This research was funded by the Federal Food Safety and Veterinary Office (FSVO), in partnership with the Federal Offices of Public Health (FOPH) and Energy (SFOE) in Switzerland, through the project LeCo (Legionella Control in Buildings; Aramis nr.:4.20.01) and Eawag discretionary funding. Data produced and analyzed in this paper were generated in collaboration with the Genetic Diversity Centre (GDC), ETH Zurich.

## AUTHOR AFFILIATIONS

[1]Department of Environmental Microbiology, Eawag, Swiss Federal Institute of Aquatic Science and Technology, Dübendorf, Switzerland
[2]Department of Environmental Systems Science, Institute of Biogeochemistry and Pollutant Dynamics, ETH Zurich, Zürich, Switzerland

## AUTHOR ORCIDs

Alessio Cavallaro  http://orcid.org/0000-0003-4087-8216

Frederik Hammes ⓘ http://orcid.org/0000-0002-4596-4917

## FUNDING

| Funder | Grant(s) | Author(s) |
|---|---|---|
| Bundesamt für Lebensmittelsicherheit und Veterinärwesen (FSVO) | LeCo (Legionella Control in Buildings Aramis nr.:4.20.01) | Alessio Cavallaro |
| Bundesamt für Gesundheit (BAG) | LeCo (Legionella Control in Buildings Aramis nr.:4.20.01) | Alessio Cavallaro |
| Bundesamt für Energie (BFE) | LeCo (Legionella Control in Buildings Aramis nr.:4.20.01) | Alessio Cavallaro |
| Eawag, Swiss Federal Institute of Aquatic Science and Technology | | Marco Gabrielli |
| | | Frederik Hammes |
| | | William J. Rhoads |

## DATA AVAILABILITY

The data for this study have been deposited in the European Nucleotide Archive (ENA) at EMBL-EBI under accession number PRJEB72629. All other data sets from this study are available from the Eawag Research Data Institutional Repository.

## ADDITIONAL FILES

The following material is available online.

### Supplemental Material

**Supplemental figures and tables (Spectrum00713-24-s0001.docx).** Fig. S1-S6; Tables S1 and S2.
**Supplemental material (Spectrum00713-24-s0002.txt).** Reports for the initial data preparation conducted on the 16S amplicon sequencing raw data.

### Open Peer Review

**PEER REVIEW HISTORY (review-history.pdf).** An accounting of the reviewer comments and feedback.

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
