## [Reviewer comments · Microbiology Spectrum]

Microbiology Spectrum

The impact of DNA extraction on the quantification of *Legionella*, with implications for ecological studies

Alessio Cavallaro, Marco Gabrielli, Frederik Hammes, and William Rhoads

Corresponding Author(s): Frederik Hammes, Eawag

Review Timeline:

Submission Date:	March 20, 2024
Editorial Decision:	May 9, 2024
Revision Received:	May 28, 2024
Accepted:	June 6, 2024

Editor: Sébastien Faucher

Reviewer(s): Disclosure of reviewer identity is with reference to reviewer comments included in decision letter(s). The following individuals involved in review of your submission have agreed to reveal their identity: Maria Luisa Ricci (Reviewer #1); Francesca Pennino (Reviewer #2)

Transaction Report:

DOI: <https://doi.org/10.1128/spectrum.00713-24>

Re: Spectrum00713-24 (The impact of DNA extraction on the quantification of Legionella, with implications for ecological studies)

Dear Dr. Frederik Hammes:

Thank you for the privilege of reviewing your work. Below you will find my comments, instructions from the Spectrum editorial office, and the reviewer comments.

Your manuscript was reviewed by two experts in the field. Both found your manuscript technically sound and interesting and suggested a few minor corrections. Please note that reviewer 1's comments are included in the attached file. In addition, I would suggest that for the NMDS analysis outlier samples could be included to give a sense of scale. This could be shown in a separate figure (e.g. figure 4B). A set of samples coming from another type of engineered water systems or from a different country, for example, could be used. I suggest that you try it and decide if it is worth including it.

Revision Guidelines

Sincerely,
Sébastien Faucher
Editor
Microbiology Spectrum

Reviewer #1 (Comments for the Author):

no comments

Reviewer #2 (Comments for the Author):

Dear Editor,

in the manuscript titled "The impact of DNA extraction on the quantification of Legionella, with implications for ecological studies" the Authors aimed to quantify the impact that DNA extraction methods had on downstream PCR quantification and community sequencing.

It's a very interesting topic and gives the opportunity to reflect on the importance of the decision of the Legionella detection method. Furthermore, materials and methods and results section are well described.

My final opinion is to accept the manuscript with minor revisions.

- 1) At Line 33 of the "Importance" section, you haven't added the entire name of ddPCR but only the acronym. Since this is the first time that you mention this word, you should add the entire name and from the second time you can write only the acronym.
- 2) In the "Introduction" section, you could write some numbers about Legionella infection in your country (if they are available).
- 3) In the "Materials and methods" section, can you write brand, city and country of every material that you have mentioned for the analysis? (IDEXX's Legiolert liquid culture kit at Line 115, polycarbonate filters at Line 117, etc.).
- 4) You should consult the website of the journal where you want to publish the paper on how the numbers in thousands should be written. Sometimes you put a comma (1,000), sometimes not (1000). All numbers should be written in the same way and it may vary from journal to journal.
- 5) At Line 215 of the "Data analysis" section, the parenthesis hasn't been closed after "(version 1.42.0".
- 6) At Line 268 of the "Results" section, there is a closed parenthesis but there isn't its opening.
- 7) At Line 271-272 of the "Results" section, can you better write the sentence "As expected, there instances when both methods detected L. pneumophila DNA but the cultivation data was negative".
- 8) At Line 290 of the "Results" section, you should write "(a)" in bold, as you did for (b) and (c).
- 9) At Line 316 of the "Results" section, you should write "Segata et al." as "Segata et al.", as you did for the entire text for the other references.
- 10) At Line 318 of the "Results" section, there is an extra space between the words "5. Based".
- 11) At Line 374 of the "Discussion" section, the closed parenthesis is missing after "(Fig. 2".
- 12) At Line 529 of the "Discussion" section, you should write the entire name of CFU (although its meaning is widely known), since this is the first time that you mention it.
- 13) In the "Discussion" section, you could write some examples of studies that have investigated Legionella presence in environmental samples. Don't forget that there are a lot of studies based on culture media methods, such as <https://doi.org/10.3390/ijerph20085526>.

The manuscript: The impact of DNA extraction on the quantification of Legionella, with implications for ecological studies by Alessio Cavallaro, Marco Gabrielli, Frederik Hammes, William J. Rhoads'

it is a valuable and interesting study that provides a great contribution to the detection of Legionella in water samples. As mentioned by the authors, an efficient DNA extraction is a crucial step for risk assessment analysis but also for understanding the ecology of Legionella within water and biofilm microbial communities.

Remarks

2. Material and methods

Please describe the two methodologies A and B better in the text in order to make it clearer to the reader the link to the Results section where he finds the methods A and B which are not so clearly indicated in the methods. So please specify Method A consists of; Method B consists of:.

References

The reference Scaturro et al. must be updated. Please find the exact reference at Frontiers in Microbiome: <https://www.frontiersin.org/articles/10.3389/frmbi.2023.1170824/full>

The manuscript: The impact of DNA extraction on the quantification of Legionella, with implications for ecological studies by Alessio Cavallaro, Marco Gabrielli, Frederik Hammes, William J. Rhoads'

it is a valuable and interesting study that provides a great contribution to the detection of Legionella in water samples. As mentioned by the authors, an efficient DNA extraction is a crucial step for risk assessment analysis but also for understanding the ecology of Legionella within water and biofilm microbial communities.

Remarks

2. Material and methods

Please describe the two methodologies A and B better in the text in order to make it clearer to the reader the link to the Results section where he finds the methods A and B which are not so clearly indicated in the methods. So please specify Method A consists of; Method B consists of:.

References

The reference Scaturro et al. must be updated. Please find the exact reference at Frontiers in Microbiome: <https://www.frontiersin.org/articles/10.3389/frmbi.2023.1170824/full>

RESPONSE TO REVIEWERS for

The impact of DNA extraction on the quantification of *Legionella*, with implications for ecological studies.

Alessio Cavallaro, Marco Gabrielli, Frederik Hammes, William J. Rhoads

Editor: "In addition, I would suggest that for the NMDS analysis outlier samples could be included to give a sense of scale. This could be shown in a separate figure (e.g. figure 4B). A set of samples coming from another type of engineered water systems or from a different country, for example, could be used. I suggest that you try it and decide if it is worth including it."

Authors: We appreciate the suggestion of the editor and recognize the value of including outlier samples to the distance analysis in order to give a sense of scale. We tried the analysis using a few samples selected from multiple other studies in order to reflect differences in geographic location, engineered water systems, chlorinated and unchlorinated systems, and sample types (biofilm and water). Due to the challenges related to the different variable regions of the 16S rRNA gene amplified in different studies, we decided to perform a distance analysis using the software mash (v2.3), which uses a MinHash approach and computes distance in the form of Jaccard index that accounts for the fractions of shared k-mers (Ondov et al. 2016, <https://doi.org/10.1186/s13059-016-0997-x>). We first tested this approach on only our samples, resulting in a significant separation between water and biofilm (PERMANOVA, p-value < 0.001). Once confirmed that the method was able to produce a similar outcome compared to the original NDMS (fig.4) with only the samples used in this study, we included the outliers and re-performed the analysis. We included the figure in the supplementary information as Figure S5 and attached it to the present document (see below).

Reviewer 1

Authors: The comments of the reviewer are much appreciated and helped the clarification of critical points in the manuscript. The changes to the revised manuscript are indicated with the appropriate line numbers.

1. Please describe the two methodologies A and B better in the text in order to make it clearer to the reader the link to the Results section where he finds the methods A and B which are not so clearly indicated in the methods. So please specify Method A consists of;; Method B consists of:.

Authors: A clearer description of the methodologies A and B have been provided in Material and Methods and the text adapted as follows:

Revised text: Line 140. "Two commercially available DNA extraction kits (hereafter indicated as Method A and Method B) with the same adaptations were used on each water and biofilm sample set. Each kit is commonly used for extracting DNA from environmental water samples. Both Method A and B are spin-column-based DNA extraction procedures, consisting of similar protocols, which involve the use of a chemical lysis solution, a beat-beating step and a binding matrix solution prior to spin-column concentration, washing via centrifugation and a

final elution in 100 µL. The specific composition of the reagents provided for each steps is not disclosed by the respective manufacturers.”

2. The reference Scaturro et al. must be updated. Please find the exact reference at Frontiers in Microbiome: <https://www.frontiersin.org/articles/10.3389/frmbi.2023.1170824/full>

Authors:

The reference has been updated accordingly.

Reviewer 2

Authors: The reviewer’s contribution is much appreciated and clarified several points in different sections of the manuscript. The changes to the revised manuscript are indicated with the appropriate line numbers.

1. At Line 33 of the "Importance" section, you haven't added the entire name of ddPCR but only the acronym. Since this is the first time that you mention this word, you should add the entire name and from the second time you can write only the acronym.

Authors: The text has been adapted as “digital droplet PCR”.

2. In the "Introduction" section, you could write some numbers about Legionella infection in your country (if they are available).

Authors: The text has been adapted as follows to include some numbers related to Legionnaire’s disease rates:

Revised text: Line 44. “The incidence of Legionnaire’s disease has been increasing worldwide in the past decades. In Switzerland, the reported incidence rate per 100,000 population was 7.24 in 2023, compared to 6.61 in 2017 and 3.54 in 2014”

3. In the "Materials and methods" section, can you write brand, city and country of every material that you have mentioned for the analysis? (IDEXX's Legiolert liquid culture kit at Line 115, polycarbonate filters at Line 117, etc.).

Authors: The "Material and methods" section has been updated to include the brand, city and country of the material used in this study, as suggested.

4. You should consult the website of the journal where you want to publish the paper on how the numbers in thousands should be written. Sometimes you put a comma (1,000), sometimes not (1000). All numbers should be written in the same way and it may vary from journal to journal.

Authors: All the numbers in thousands were changed and the style using a comma (e.g. 1,000) was adopted for consistency throughout the entire paper, as suggested.

5. At Line 215 of the "Data analysis" section, the parenthesis hasn't been closed after "(version 1.42.0".

Authors: The parenthesis has been closed, as suggested.

6. At Line 268 of the "Results" section, there is a closed parenthesis but there isn't its opening.

Authors: An open parenthesis has been added before "Wilcox paired test" at line 269.

7. At Line 271-272 of the "Results" section, can you better write the sentence "As expected, there instances when both methods detected *L. pneumophila* DNA but the cultivation data was negative".

Authors: The sentence was adapted as suggested.

Revised text: Lines 283-285. "As expected, cases were observed, in which both methods detected *L. pneumophila* DNA, while the cultivation data resulted negative."

8. At Line 290 of the "Results" section, you should write "(a)" in bold, as you did for (b) and (c).

Authors: "(a)" has been changed to bold, as suggested.

9. At Line 316 of the "Results" section, you should write "Segata et al." as "Segata et al.", as you did for the entire text for the other references.

Authors: The reference style was adapted to the rest of the manuscript, as suggested.

10. At Line 318 of the "Results" section, there is an extra space between the words "5. Based".

Authors: The space has been removed, as suggested.

11. At Line 374 of the "Discussion" section, the closed parenthesis is missing after "(Fig. 2".

Authors: The closing parenthesis has been added.

12. At Line 529 of the "Discussion" section, you should write the entire name of CFU (although its meaning is widely known), since this is the first time that you mention it.

Authors: The entire name (Colony Forming Units) has been added to the text, as suggested.

13. In the "Discussion" section, you could write some examples of studies that have investigated Legionella presence in environmental samples. Don't forget that there are a lot of studies based on culture media methods, such as <https://doi.org/10.3390/ijerph20085526>.

Authors: A mention to the environmental monitoring of Legionella (specifically for the prevention of nosocomial infections) has been added to the "Discussion" section, in 4.4.2

Revised text: Lines 543-548. Investigations of the presence of *Legionella* in environmental samples, which can be used as a surveillance strategy, are often carried out using a culture-dependent approach. For example, several studies have reported environmental monitoring of *Legionella* in hospital settings over multiple years, in order to prevent nosocomial infections and link cases of Legionnaires' disease with the environmental source of infection.

Re: Spectrum00713-24R1 (The impact of DNA extraction on the quantification of Legionella, with implications for ecological studies)

Dear Dr. Frederik Hammes:

Your manuscript has been accepted, and I am forwarding it to the ASM production staff for publication. Your paper will first be checked to make sure all elements meet the technical requirements. ASM staff will contact you if anything needs to be revised before copyediting and production can begin. Otherwise, you will be notified when your proofs are ready to be viewed.

Sincerely,
Sébastien Faucher
Editor
Microbiology Spectrum